# Impact of the COVID-19 pandemic on bone and soft tissue tumor treatment: A single-institution study

Yoshitaka Ban[ID]*, Manabu Hoshi, Naoto Oebisu, Akiyoshi Shimatani, Naoki Takada, Tadashi Iwai, Hiroaki Nakamura

Department of Orthopedic Surgery, Osaka City University Graduate School of Medicine, Osaka, Japan

* ychbanchan@gmail.com

**Data Availability Statement:** All relevant data are within the manuscript and its Supporting Information files.

## Abstract

### Objective

The spread of coronavirus disease 2019 (COVID-19) has caused a great deal of damage to daily medical care. We investigated the impact of the COVID-19 pandemic on bone and soft tissue tumor treatment at our hospital.

### Methods

We conducted a retrospective comparative study of two groups of patients at Osaka City University Hospital during the period of increasing COVID-19 infections (February-December 2020, group C) and the same period the previous year (February- December 2019, group NC). Clinical data, including patient's age, gender, type of tumor, neoplasms, number of surgical cases for inpatients and outpatients, operation time, use of implants, length of hospital stay, inpatient hospital costs, number of inpatients receiving anticancer drugs, and postoperative complications in these two groups were retrospectively evaluated.

### Results

The number of cases of malignant bone and soft tissue tumors that were resected during hospitalization was predominantly higher in group C than in group NC (P = 0.01). There were no significant differences in operation time, use of implants, and postoperative complications between group C and group NC, but there were significant differences in the length of hospital stay and hospital costs (P<0.001).

### Conclusions

The COVID-19 pandemic has been recognized throughout the world to have adverse effects in a variety of areas. It had a negative impact on hospital costs and the length of hospital stay in the field of bone and soft tissue tumor treatment.

**Funding:** The author(s) received no specific funding for this work.

## Introduction

Several mysterious pneumonia cases were reported for the first time in China (Wuhan) at the end of 2019, and severe acute respiratory syndrome coronavirus-type 2 (SARS-CoV-2) was identified as the RNA virus causing 'Coronavirus Disease 2019'(COVID-19) [1]. SARS-CoV-2 belongs to a family of coronaviruses that have single-stranded plus-stranded RNA as their genome [2]. Coronavirus includes a very large number of species that infect mammals and birds, but those that cause infections in humans are SARS coronavirus (SARS-CoV) and MERS coronavirus (MERS-CoV); they cause severe pneumonia [2]. There are two possible routes of infection, droplet infection and contact infection, and it is extremely important to take general infection control measures such as hand disinfection with alcohol and wearing a mask to prevent infection. The incubation period is reported by the World Health Organization (WHO) to be 1 to 12.5 days (mostly 5 to 6 days), and it is recommended that patients suspected to be infected be monitored for 14 days [3].

COVID-19 rapidly spread and the WHO declared it a global pandemic on 11 March 2020 [4]. At present, the total cumulative number of COVID-19 cases in the world has reached more than 545 million [5]. The economic losses are so great that various countries are providing economic support measures. In Japan, the government declared a state of emergency on 7 April 2020 [6] while implementing additional economic stimulus measures to control the infection, but the spread of the infection caused a great deal of damage to daily medical care, and there was a shortage of doctors and nurses, a shortage of hospital beds, and a near-collapse of the medical system.

Three vaccines have been shown to avail immunity against COVID-19: AstraZeneca, Moderna, and Pfizer. Of these, Pfizer's coronavirus vaccine was the first to be introduced in Japan [7]. The American pharmaceutical giant Pfizer developed the vaccine in collaboration with Biontech (the German biotech company) [8], and vaccinations began in the UK in December 2020 [9]. In Japan, the vaccine was approved on February 12, 2021, and prior vaccinations were conducted for medical personnel and the elderly aged 65 years and older, as well as people with underlying diseases [10]. All of our medical staff have completed Pfizer's vaccination program.

The COVID-19 pandemic has had a negative impact on many medical fields [11–13]. For example, in cancer treatment, the effects of COVID-19 have led to adjustments in treatments such as anticancer drugs and immunotherapy, and have caused treatment delays or discontinuation [13]. In addition, several operations in various surgical fields have been canceled or postponed due to the effects of COVID-19 [14]. However, no previous reports have described the impact of the COVID-19 pandemic on bone and soft tissue tumor treatment. In this study, we investigated the impact of the COVID-19 pandemic on bone and soft tissue tumor treatment at our hospital.

## Materials and methods

We conducted a retrospective comparative study of two groups of patients at Osaka City University Hospital during the period of increasing COVID-19 infections (February-December 2020, group C) and the same period the previous year (February- December 2019, group NC). The inclusion criteria were outpatients and inpatients who visited Osaka City University Hospital and patients treated with anticancer drugs. The exclusion criterion was having incomplete medical records. The following items were examined: patient's age, gender, type of tumor (bone or soft tissue) and neoplasms (benign or malignant), number of surgical cases (outpatient and inpatient surgery), operation time, use of implants, length of hospital stay, inpatient hospital cost, number of outpatients (inpatients), and number of inpatients receiving

anticancer drugs. In addition, perioperative complications (soft tissue necrosis, pneumothorax, sepsis, wound infection, fracture, etc) were recorded and defined as grade ≥3 based on the Common Terminology Criteria for Adverse Events (CTCAE v5.0; National Cancer Institute). The anticancer drugs for malignant bone and soft tissue tumors were as follows: For the anticancer treatment of malignant bone and soft tissue tumors, we used a method similar to that reported by Hoshi et al [15]. Drugs for anticancer therapy for osteosarcoma included adriamycin (DOX) and Cisplatin (CDDP) as well as ifosfamide (IFM) and methotrexate (MTX). Drugs for anticancer therapy for Ewing sarcoma included vincristine (VCR) and DOX as well as cyclophosphamide alternating with IFM, and etoposide (ETP) repeating every 21 days.

Patients with high-grade soft tissue sarcomas received DOX and IFM-based chemotherapy. Patients with metastatic soft tissue sarcomas received gemcitabine (GEM) and the combination with docetaxel (DTX). Patients who had soft tissue sarcomas refractory to, or who could not tolerate, anthracycline-based chemotherapy received trabectedin as second-line chemotherapy. Patients who had advanced or metastatic soft tissue sarcomas received eribulin as second-line chemotherapy.

For patients with relapse in the lungs or other sites, second-line chemotherapy comprised IFM, carboplatin, and ETP, with some modifications.

Anticancer drug-related adverse events (such as decreased neutrophil count, febrile neutropenia, decreased platelet count, pneumothorax, disseminated intravascular coagulation, pleural effusion, and heart failure) were also investigated and defined as grade ≥3 based on the CTCAE.

This retrospective study was approved by the Institutional Review Board of Osaka City University Graduate School of Medicine and was performed in accordance with the ethical standards laid down in the Declaration of Helsinki (no. 4934). The subjects included in the study provided informed consent prior to their inclusion in the study.

### Statistical analysis

Fisher's exact probability test was performed for analyzing categorical variables (use of implants, anticancer drug-related adverse events, and postoperative complications). The Mann–Whitney U test was used to compare various clinical factors. Values of $P < 0.05$ were considered statistically significant. Statistical analysis was performed using the Excel Statistics software for Windows (version 2020; SSRI Co., Ltd., Tokyo, Japan)

## Results

### Clinical characteristics of the two groups

Table 1 shows a comparison of the clinical characteristics of group C Vs. group NC for inpatients. Of the 253 patients treated for bone and soft tissue tumors, 114 (45%) patients were in group C while 139 (55%) patients were in group NC. The patients' mean age was 57±21 years and 66 (58%) were male in group C, and in group NC, the mean age was 55±22 years and 78 (56%) were male.

There were 44 cases in which the site of tumor origin was a bone tumor in group C and 59 cases in group NC. There were 70 cases in which the site of tumor origin was a soft tissue tumor in group C and 80 cases in group NC. Histologically malignant cases were found in 57 cases in group C and in 50 cases in group NC. Histologically benign cases were found in 57 cases in group C and in 89 cases in group NC. The number of inpatients in group C was higher than that in group NC, while the number of surgeries for malignant bone and soft tissue tumors was higher than in group NC (P = 0.01). The median (range) operation time was 70 (42–138) min and 56 (33–97) min in groups C and NC, respectively (P = 0.2). Sixteen cases

**Table 1. Comparison of clinical information of corona era vs no corona era for inpatients.**

| Characteristics | Group C | Group NC | P value |
|---|---|---|---|
| | N = 114 | N = 139 | |
| Median age (years) | 57 (7–92) | 55 (5–88) | 0.7 |
| Gender | | | |
| Male | 66 | 78 | 0.4 |
| Female | 48 | 61 | |
| Type of tumor | | | |
| Bone | 44 | 59 | 0.3 |
| Soft tissue | 70 | 80 | |
| Neoplasms | | | |
| Malignant | 57 | 50 | 0.01 |
| Benign | 57 | 89 | |
| Operation time (min) | 70 (13–351) | 56 (15–515) | 0.2 |
| Use of Implant | | | |
| Yes | 16 | 16 | 0.3 |
| No | 98 | 123 | |
| Postoperative complications (CTCAE v5.0) | 8 | 4 | 0.15 |
| Soft tissue necrosis | 5 | 2 | |
| Pneumothorax | 1 | | |
| Sepsis | 1 | | |
| Wound infection | 1 | 1 | |
| Fracture | | 1 | |
| Median length of hospital stay (days) | 12 (2–89) | 8 (2–117) | <0.001 |
| Hospital cost (US dollars) | 8,497 | 5,305 | <0.001 |

underwent surgery with implants in both groups C and NC (P = 0.3). Eight cases had postoperative complications, such as soft tissue necrosis, wound infection, pneumothorax, and fracture, in group C and 4 cases in group NC (P = 0.15). There were no significant differences in operation time, use of implants, and postoperative complications between groups C and NC. The median length of hospital stay was 12 (8–23) days in group C and 8 (5–16) days in group NC. The median hospital cost was USD 8497 (4733–13845) and 5305 (3598–9197) in groups C and NC, respectively. There were significant differences in the median length of hospital stay and the median hospital cost between groups C and NC (P<0.001).

Table 2 shows a comparison of the clinical characteristics of group C VS those of group NC for day surgery and chemotherapy. The number of cases of day surgery was 16 in group C and 26 in group NC. We also surveyed the number of patients who visited the outpatient clinic of orthopedic surgeons who have sub-specialized in bone and soft tissue tumor care at our hospital.

Table 3 shows a comparison of the number of outpatients in the COVID-19 era VS the pre-COVID-19 era. The total numbers of outpatients who visited the outpatient clinic of bone and soft tissue oncologists were 2429 in group C and 2548 in group NC. By month, the increase in the number of COVID-19-positive patients had little impact on the number of outpatients in 2020.

In this study, we administered various anticancer drugs to patients, and ADR, IFM, and trabectedin were the most commonly used drugs among anticancer drugs. Neutrophil count decrease was the most common type of anticancer drug-related adverse event. There were significant differences in anticancer drug-related adverse events between groups C and NC (P = 0.03).

**Table 2. Comparison of clinical information of corona era vs no corona era for day surgery and chemotherapy.**

| Characteristics | Group C | Group NC | P value |
|---|---|---|---|
| Day surgery | 16 | 26 | 0.4 |
| Malignant | 3 | 3 | |
| Benign | 13 | 23 | |
| Chemotherapy | 90 | 64 | |
| DOX, IFM | 18 | 14 | |
| DOX | 3 | 3 | |
| IFM | 4 | 6 | |
| Trabectedin | 21 | 12 | |
| IE | 12 | 7 | |
| VDC | 13 | 6 | |
| DOX CDDP | 11 | 1 | |
| ICE | 4 | 2 | |
| MTX | 2 | 4 | |
| GEM DTX | 1 | 6 | |
| Eribulin | 1 | 3 | |
| Anticancer drug-related adverse events | 14 | 20 | 0.03 |
| Neutrophil count decreased | 10 | 13 | |
| Febrile neutropenia | 2 | 2 | |
| Platelet count decreased | 1 | 2 | |
| Pneumothorax | | 1 | |
| Disseminated intravascular coagulation | | 1 | |
| Pleural effusion | | 1 | |
| Heart failure | 1 | | |

DOX, adriamycin; IFM, ifosfamide; VDC, vincristine and cyclophosphamide and actinomycin-D

CDDP, cisplatin; ICE, ifosfamide and cisplatin and etoposide; MTX, methotrexate; GEM, gemcitabine

DTX, docetaxel

**Table 3. Comparison of number of outpatients in corona era vs no corona era.**

| Month | Group C | Group NC | Number of new PCR positive patients in 2020 | P value |
|---|---|---|---|---|
| | N = 2429 | N = 2548 | | |
| January | 223 | 323 | 1 | |
| February | 173 | 205 | 0 | |
| March | 254 | 253 | 14 | |
| April | 208 | 188 | 257 | |
| May | 122 | 182 | 282 | |
| June | 204 | 188 | 36 | |
| July | 263 | 262 | 125 | |
| August | 201 | 196 | 1537 | |
| September | 200 | 145 | 624 | |
| October | 205 | 213 | 619 | |
| November | 169 | 176 | 603 | |
| December | 207 | 217 | 2014 | |
| Total | 2429 | 2548 | | 0.9 |

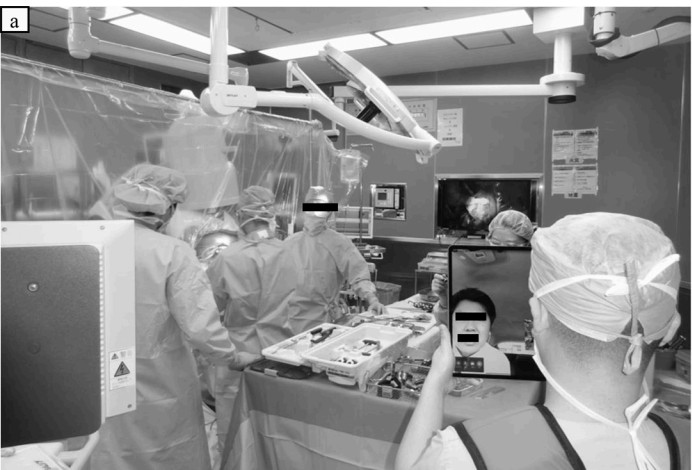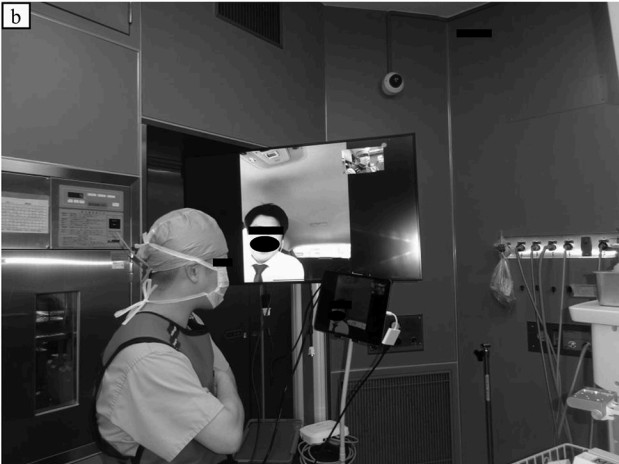

**Fig 1. A vendor attending a surgery online.** Images of a vendor attending a surgery online.

In orthopedic surgery, implants are often used, and the surgery is performed in the presence of a vendor. However, with the outbreak of COVID-19, our hospital director took measures to prohibit the presence of vendors and reduce the number of people in the operating room as much as possible to prevent the spread of COVID-19. We thus changed to online attendance in orthopedic surgery by adopting remote work. Fig 1 shows that we performed surgery for a left femur bone tumor with curettage and plate fixation using Compression Hip Screw.

The use of a tablet allowed us to see each other's images, and we could directly see with our eyes whether there were any mistakes in the implant insertion procedure or other processes. It was also possible to respond immediately to implant problems during surgery. However, there are many surgical instruments in orthopedic surgery, in addition to the difficulty of direct explanation and guidance from the contractor when online, so the surgeon and the operating nurse had to be familiar with the implant.

## Discussion

The global coronavirus pandemic has had a profoundly negative impact on the healthcare industry [11]. In addition to the shortage of medical supplies [16], hospital beds [12], and personnel [17], the pandemic has also affected the surgical field [18–21] and caused the postponement of scheduled surgeries [14].

In this study, the number of inpatient and outpatient surgeries showed a downward trend compared to the numbers in the pre-COVID-19 era, but the number of surgeries for malignant bone and soft tissue tumors for inpatients remained the same compared to the number the previous year. Several reports have shown that the COVID-19 pandemic has contributed to a decrease in the number of surgeries for various carcinomas [18, 20, 21]; conversely, other studies have reported that the number of tumor surgeries has increased [22], and the impact of the COVID-19 pandemic on surgery is controversial. In our hospital, surgeries for malignant bone and soft tissue tumors were actively performed. The French Sarcoma group mentions that it is not recommended to delay surgery for operable patients without COVID symptoms, in particular for grade 2 or 3 soft tissue sarcoma [23]. Surgical treatment of the primary tumor is inevitable for the treatment of malignant bone and soft tissue tumors. Malignant bone and soft tissue tumors grow at a rapid rate, and the larger the tumor, the more difficult it is to resect

a large area, making local control of the tumor difficult. Regardless of histologic grade, the prognosis for locally recurrent tumors is poor. Therefore, early surgical treatment is necessary for patients without specific symptoms of coronavirus and should take precedence over surgery for degenerative diseases.

A previous report on the association between anticancer drugs and COVID-19 speculated that people receiving systemic anticancer therapy have a higher risk of death from COVID-19, especially for hematological malignancies, but there was no significant difference in the solid tumor group [24]. In this study, there were significant differences in the anticancer drug-related adverse events between groups C and NC (P = 0.03). However, it is unlikely that COVID-19 is associated with anticancer drug-related adverse events because of the large number of anticancer drugs used in this study and the variety of patient backgrounds.

All patients admitted to the hospital underwent a Polymerase Chain Reaction (PCR) test before admission, and in principle, patients were admitted after confirming that the PCR test was negative. As a countermeasure against COVID-19 in hospitalized patients undergoing anticancer drug treatment, all patients with fever were placed in a private room and their physician was equipped with personal protective equipment; the patients also underwent PCR testing again. If the PCR test was negative, we suspected the possibility of febrile neutropenia, submitted various culture tests, and empirically administered antibiotics. If the PCR test was positive, the patient was judged to be COVID-19-positive, and the patient's general condition was carefully monitored with continued private room management.

In this study, the median length of hospital stay was 12 (8–23) days in group C and 8 (5–16) days in group NC. The median hospital costs were USD 8,497 (4,733–13,845) and 5,305 (3,598–9,197) in groups C and NC, respectively. There were significant differences in the median length of hospital stay and the median hospital cost between groups C and NC (P<0.001). The COVID-19 pandemic has forced hospitals to take a variety of measures. At our hospital, patients were often hospitalized the day before surgery before the coronavirus pandemic. During the COVID-19 pandemic, however, patients were hospitalized several days before surgery to confirm the absence of fever and respiratory symptoms, and PCR testing was also performed on the day of admission. Therefore, the need to conduct PCR tests and chest computed tomography after the coronavirus outbreak to rule out pneumonia may have been a cause for the increased hospital costs and prolonged length of stay. These measures may have contributed to the increase in the length of hospital stay and the rising cost of medical care.

Japan has become a super-aged society, with the elderly population accounting for 28.6% of the total population: one of every four people in Japan is aged 65 years or older; this is a record high [25]. As a result, medical costs are rapidly rising, and it is necessary to reduce medical costs in the future. However, the large outbreak of COVID-19 may cause a huge loss to the Japanese economy due to medical costs.

In this study, we investigated the impact of the COVID-19 pandemic on the bone and soft tissue tumor field. There was no change in the number of inpatient surgeries for malignant bone and soft tissue tumors, and only the number of surgeries for benign tumors decreased, while there was no change in the number of surgeries for outpatients. The addition of tests, such as PCR and preoperative chest CT, at the time of hospitalization may lead to longer hospital stays and higher medical costs. In addition, the number of outpatients did not fluctuate according to the number of COVID-19 cases.

The cancer surgery system worldwide has been found to be fragile to lockdowns [18], and it is important to better monitor changes in the quality of care, identify the causes of these changes, and take a variety of measures accordingly since these global healthcare crises are expected to occur in the future. Previous reports have shown a minimal impact of COVID-19, largely due to the decision to promptly halt all elective surgical procedures and deferrable

outpatient care [26]. In addition, there are reports that the complications of postoperative pneumonia were reduced by creating a dedicated COVID-19-free pathway in the operating room, ward, and intensive care unit [27]. In our hospital, patients who have developed COVID-19 are transported separately from the normal route, and an isolation ward is set up in our hospital. In addition, as shown in Fig 1, we respond to such medical crises by making full use of information technology during surgery to minimize the number of people coming in and out of the hospital as much as possible.

## Conclusion

We conducted a study on the impact of the COVID-19 pandemic on bone and soft tissue tumor treatment and found that the COVID-19 pandemic had an impact on the cost of medical care, but no impact on surgery for malignant bone and soft tissue tumors. The COVID-19 pandemic caused a near-collapse of the healthcare system, including a shortage of medical resources, medical personnel, and hospital beds. The COVID-19 pandemic had an impact on hospital costs and the length of hospital stay in the field of orthopedics.

## Supporting information

**S1 File. All data.**
(PDF)

## Acknowledgments

The authors are very grateful for the invaluable support and various discussions with other members of the Department of Orthopedic Surgery.

## Author Contributions

**Conceptualization:** Manabu Hoshi.

**Data curation:** Yoshitaka Ban, Naoto Oebisu, Akiyoshi Shimatani, Naoki Takada, Tadashi Iwai.

**Formal analysis:** Yoshitaka Ban.

**Investigation:** Yoshitaka Ban, Akiyoshi Shimatani, Naoki Takada, Tadashi Iwai.

**Methodology:** Yoshitaka Ban.

**Project administration:** Manabu Hoshi.

**Software:** Yoshitaka Ban.

**Supervision:** Manabu Hoshi, Naoto Oebisu, Hiroaki Nakamura.

**Visualization:** Manabu Hoshi.

**Writing – original draft:** Yoshitaka Ban.

**Writing – review & editing:** Manabu Hoshi.

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
