## [Decision Letter · Decision Letter 0]

2 Jan 2023

PONE-D-22-31176Impact of the COVID-19 pandemic on bone and soft tissue tumor treatment: A single-institution studyPLOS ONE

Dear Dr. Ban,

Thank you for submitting your manuscript to PLOS ONE. After careful consideration, we feel that it has merit but does not fully meet PLOS ONE’s publication criteria as it currently stands. Therefore, we invite you to submit a revised version of the manuscript that addresses the points raised during the review process.

We look forward to receiving your revised manuscript.

Kind regards,

Fulvio Borella

Academic Editor

PLOS ONE

Journal Requirements:

Additional Editor Comments:

Minor revision

Reviewers' comments:

Reviewer's Responses to Questions

**Comments to the Author**

1. Is the manuscript technically sound, and do the data support the conclusions?

Reviewer #1: Yes

Reviewer #2: Yes

2. Has the statistical analysis been performed appropriately and rigorously? 

Reviewer #1: Yes

Reviewer #2: Yes

3. Have the authors made all data underlying the findings in their manuscript fully available?

Reviewer #1: Yes

Reviewer #2: Yes

4. Is the manuscript presented in an intelligible fashion and written in standard English?

Reviewer #1: Yes

Reviewer #2: Yes

5. Review Comments to the Author

Reviewer #1: The manuscript entitled "Impact of the COVID-19 pandemic on bone and soft tissue tumor treatment: A single-institution study" by Yoshitaka Ban and co-authors addresses a common and widely-known problem of COVID-19 pandemic and patient treatments. Here, the authors retrospectively analyzed their clinical performance regarding oncological therapy of patients suffering from bone and soft tissue tumors. The authors could show that there were mainly no differences in treatment during COVID-19 compared to no-Covid-19 period except that significantly more patients have been operated during hospitalization. Furthermore the authors show that length of hospital stay was longer and hospital costs were higher.

The manuscript is well written and structured, however, there are some issues that should be a addressed before a recommendation for publication can be given:

1. Please show the results in table 1 that more patients have been operated during hospitalization during COVID-19 pandemic.

2. Please comment on this point why more patients have been operated during hospitalization!

3. Are the differences of of LOS and costs are the result that more patients have been operated during hospitalization? Please comment on that.

4. Please add more actual and important references such as PMID: 34624250, PMID: 33021869.

Reviewer #2: The article is interesting, well written and analyzes the management problem of rare tumors during the COVID-19 pandemic

I would just suggest that you improved the discussion by comparing your data with other single-center experiences or other reports.

Please consider these papers to spice up the discussion:

https://doi.org/10.1016/j.suronc.2022.101859

10.7759/cureus.30785

https://doi.org/10.3390/healthcare10112329

https://doi.org/10.1186/s12913-022-08166-0

https://doi.org/10.1002/jso.26256

6. PLOS authors have the option to publish the peer review history of their article (what does this mean?). If published, this will include your full peer review and any attached files.

Reviewer #1: **Yes: **Sven Flemming

Reviewer #2: No

---

## [Author Response · Author response to Decision Letter 0]

2 Feb 2023

We thank the editor for these suggestions.

Reviewer 1: I have incorporated all of your suggestions into my revision. They were very helpful. Thank you.

Reviewer 2: I have incorporated all of your suggestions into my revision. They were very helpful. Thank you.

---

## [Editor Report · Decision Letter 1]

20 Mar 2023

Impact of the COVID-19 pandemic on bone and soft tissue tumor treatment: A single-institution study

PONE-D-22-31176R1

We’re pleased to inform you that your manuscript has been judged scientifically suitable for publication and will be formally accepted for publication once it meets all outstanding technical requirements.

Kind regards,

Fulvio Borella

Academic Editor

PLOS ONE

---

## [Editor Report · Acceptance letter]

14 Apr 2023

PONE-D-22-31176R1 

Impact of the COVID-19 pandemic on bone and soft tissue tumor treatment: A single-institution study 

Dear Dr. Ban:

I'm pleased to inform you that your manuscript has been deemed suitable for publication in PLOS ONE. Congratulations! Your manuscript is now with our production department. 

Kind regards, 

on behalf of

Dr. Fulvio Borella 

Academic Editor

PLOS ONE